# Melanoma Brain Metastases: A Retrospective Analysis of Prognostic Factors and Efficacy of Multimodal Therapies

**DOI:** 10.3390/cancers15051542

**Published:** 2023-02-28

**Authors:** Valeria Internò, Maria Chiara Sergi, Maria Elvira Metta, Michele Guida, Paolo Trerotoli, Sabino Strippoli, Salvatore Circelli, Camillo Porta, Marco Tucci

**Affiliations:** 1Department of Interdisciplinary Medicine, University of Bari Aldo Moro, 70121 Bari, Italy; 2Medical Oncology Unit, Azienda Ospedaliero-Universitaria, Policlinico di Bari, 70124 Bari, Italy; 3Medical Statistic and Biometry Unit, Department of Biomedical Sciences and Human Oncology, University of Bari Aldo Moro, 70121 Bari, Italy; 4IRCCS, Istituto Tumori Giovanni Paolo II, 70124 Bari, Italy; 5Medical Statistic and Biometry Unit, Department of Interdisciplinary Medicine, University of Bari Aldo Moro, 70124 Bari, Italy

**Keywords:** cutaneous melanoma, brain metastases, radiotherapy, immunotherapy, melanoma brain metastases, brain metastases treatment

## Abstract

**Simple Summary:**

The treatment strategies of patients with melanoma brain metastases are continually evolving, although this remains a poor prognostic subset. We report a real-life retrospective analysis of 105 patients with melanoma brain metastases aiming to analyze the impact of clinical–pathological features and multimodal therapies, such as neurological symptom occurrence, on overall survival in the pre-combined immunotherapy era. We observed a significant improvement in the survival of patients treated with encephalic radiotherapy (eRT) despite the type of systemic treatment performed. The only subset of patients that did not experience survival improvement from eRT was identified by LDH levels higher than two times the upper limit normal. In our opinion, our results, if confirmed by prospective analysis, may help to identify the correct therapeutic strategy for the worst prognostic subgroup of patients with melanoma brain metastases.

**Abstract:**

Brain metastasis in cutaneous melanoma (CM) has historically been considered to be a dismal prognostic feature, although recent evidence has highlighted the intracranial activity of combined immunotherapy (IT). Herein, we completed a retrospective study to investigate the impact of clinical–pathological features and multimodal therapies on the overall survival (OS) of CM patients with brain metastases. A total of 105 patients were evaluated. Nearly half of the patients developed neurological symptoms leading to a negative prognosis (*p* = 0.0374). Both symptomatic and asymptomatic patients benefited from encephalic radiotherapy (eRT) (*p* = 0.0234 and *p* = 0.011). Lactate dehydrogenase (LDH) levels two times higher than the upper limit normal (ULN) at the time of brain metastasis onset was associated with poor prognosis (*p* = 0.0452) and identified those patients who did not benefit from eRT. Additionally, the poor prognostic role of LDH levels was confirmed in patients treated with targeted therapy (TT) (*p* = 0.0015) concerning those who received immunotherapy (IT) (*p* = 0.16). Based on these results, LDH levels higher than two times the ULN at the time of the encephalic progression identify those patients with a poor prognosis who did not benefit from eRT. The negative prognostic role of LDH levels on eRT observed in our study will require prospective evaluations.

## 1. Introduction

Brain metastases (BMs) occur in almost 50% of patients with cutaneous melanoma (CM) and are the third most common metastatic site [1,2,3,4]. These patients commonly experience a dismal prognosis with a 4 month median overall survival (mOS) [5,6]. Immune checkpoint inhibitors (ICIs) and targeted therapies (TTs) have improved the outcome of unresectable and advanced CM but, in the recent past, patients with active BMs were routinely excluded from clinical trials. Therefore, the evaluation of the intracranial efficacy of systemic therapies was mostly based on retrospective analyses [7,8,9,10]. More recently, however, prospective studies have been designed to include patients with BMs [8,9,11,12,13]. In particular, the COMBI-MB Phase 2 study explored the impact of Dabrafenib plus Trametinib in BRAF v600-mutant melanoma brain metastases and revealed an intracranial response rate of 58% in the entire cohort. However, the duration of response was at least 6.5 months, and was significantly lower with respect to patients without BMs resulting in at least 12 months [14,15,16]. Both anti-PD1 and anti-CTLA-4 monoclonal antibodies showed poor efficacy for the treatment of BMs when administered as single agents, thus showing a 25% intracranial response rate [17,18,19], whereas the best response was obtained by their combination. The Australian anti-PD-1 brain collaboration (ABC) study showed a rate of response of about 50% with a median duration of response not reached after 34 months [20]. In addition, the CheckMate-204 trial demonstrated an OS rate of 72% after a 3 year follow-up in untreated and asymptomatic patients while a similar benefit was not reached in those who were symptomatic and showed an intracranial response of 22% and a survival rate of 33% at 36 months of follow-up [21]. Nowadays, many critical issues emerge in the choice of the best treatment for patients with BM, and the best strategy is still debated for those bearing the BRAF mutation. In this regard, preliminary data from the Phase 2 TRICOTEL study underlined the intracranial efficacy of combining TT with ICIs, with particular activity in patients receiving corticosteroids and/or in symptomatic ones [22]. Moreover, recent evidence has proved an early acquired resistance to the combination of ICIs for patients pre-treated with TT [23]. Furthermore, growing interest has emerged in the combination of ICIs with encephalic radiotherapy (eRT) to maximize the antitumor response [24,25,26,27,28,29]. Thus, trials evaluating the putative additive effect of eRT have been designed, but results are not still available (NCT03340129, NCT03430947 and NCT02097732). Apart from neurological symptoms, other factors are questioned to have a prognostic role in brain metastatic melanoma, such as the presence of more than three BMs, a poor performance status and the concomitant presence of extracranial metastases in specific sites, as well as elevated lactate dehydrogenase (LDH) levels at the time of encephalic progression. However, the role of these parameters needs to be further confirmed [30,31,32,33,34]. Herein, we performed a retrospective, real-life, multicentric analysis to explore the prognostic role of clinical, demographic and pathological features and the efficacy of multimodal therapeutic strategies in our cohort of CM patients with BMs.

## 2. Materials and Methods

### 2.1. Study Population

This is a retrospective observational study that enrolled 105 patients with histological CM and radiological diagnosis of BMs treated in two oncological centers of Bari (Medical Oncology Unit, Policlinico Hospital, and Rare Tumors and Melanoma Unit, IRCCS Istituto Tumori Giovanni Paolo II). This study was designed and performed by medical oncologists. Patients received standard treatments according to good clinical practice. Demographic data included histopathological parameters according to AJCC Eighth Edition (e.g., histotype, Breslow depth, ulceration, number of mitoses and lymphocyte infiltration), clinical features (such as neurological symptoms), access and type of systemic or local treatments (single-agent IT, TT with anti-BRAF plus anti-MEK drugs, chemotherapy and eRT performed as stereotactic radiosurgery or whole brain radiotherapy) and time of intra- and extra-cranial metastatic diagnosis and death. Other parameters were age, sex, melanoma primary site, nodal involvement, mutational status of both BRAF and NRAS analyzed on both primary and metastatic specimens, LDH levels, sites of extracranial metastases and leptomeningeal neoplastic infiltration. Moreover, detailed information regarding systemic treatments and sequences, as well as the time of extra-cranial and intra-cranial progression, were assessed. Written informed consent for clinical data collection was collected from all patients.

### 2.2. Statistical Analysis

A comparison of cumulative survival was performed using Kaplan–Meier curves for each variable (symptoms, radiotherapy, type of systemic therapy and LDH levels). Median overall survival (mOS) and its interquartile range (IQR) were determined in each subgroup, aided by the time of BM diagnosis. Median progression-free survival (mPFS) and its IQR were determined in each subgroup, aided by the date of BM diagnosis until the first intracranial progression according to response assessment in neuro-oncology (RANO) criteria. However, due to the retrospective nature of our study and the lack of a centralized radiological evaluation, we decided to consider only mOS for the final evaluation of treatment efficacy. The difference in survival was evaluated via the log-rank test run for each variable and in the other two analyses: for eRT adjusted accounting for symptoms, and for eRT adjusted accounting for systemic therapy and for LDH levels adjusted accounting for eRT. Furthermore, the Cox regression model was applied to evaluate the effect of each variable on the risk of death. All risk factors were evaluated to assess the assumption of a proportional hazard using a multivariable Cox model with the dependent OS. The model-independent variables were as follows: treatment (TT, chemotherapy and IT), eRT (yes/no), site of the primary melanoma (head and neck, trunk, limbs and other sites), neurological symptoms (presence/absence), Breslow (≤1 mm, 1–2 mm, 2.1–4 mm and >4 mm), age classes (the classes were lower than 56 years, 56–65 years, 66–75 years and more than 75) and sex (male = 1, female = 0). All of the analyses were performed using SAS 9.4 for a personal computer via PROC LIFE TEST and PROC PHREG. Statistical significance was set at *p* < 0.05.

### 2.3. Ethics Approval and Consent to Participate

Ethical review and approval were not required for this study due to its retrospective and observational nature and due to being conducted in accordance with national regulations that strictly impose ethical review and approval only for those observational studies designed as prospective pharmacological observational ones (Official Gazette of the Italian Republic n. 42, 19 February 2022, decree 30 November 2021, art. 6 subparagraph 2). The clinical data collection on human participants is in strict compliance with the ethical standards of the Declaration of Helsinki. All patients provided their written informed consent to participate in the study and to the publication of their data in an anonymous form.

## 3. Results

### 3.1. Baseline Demographic Features

Cutaneous melanoma patients with BM were enrolled from 2017 to 2021. The clinical and pathological features are described in Table 1.

About 90% of patients (95/105) also had extracranial metastases and 39% of them were diagnosed with more than two extracranial metastatic sites. The onset of BMs was mostly metachronous in 97% of patients after the melanoma diagnosis and 61% after an extracranial progression. Therefore, the spread of melanoma toward the brain mostly occurred in a delayed phase of the disease. The brain metastatic location was supratentorial in 51% of cases and supra- and infra-tentorial in 24% of patients, while infra-tentorial occurred in 6% of patients. About 59% (62/105) of patients showed less than four MRI-confirmed BMs. At the onset of BMs, 18% of patients had LDH levels higher than 2-fold the upper level normal (ULN), whereas 45% suffered from neurological symptoms (47/105) and all of them received steroids. Neurosurgical exeresis was performed on 13 patients (12%) due to emergency reasons or therapeutic strategy. Encephalic RT was completed in 71% of patients, stereotactic radiosurgery (SRS) was performed in 65% of patients and whole brain radiotherapy (WBRT) was performed for 35% of them. With regard to first-line systemic treatments after BM diagnosis, 15 patients did not receive any therapy due to their poor performance status. Among the others (n = 90), 48% received TT and 48% received IT, whereas only 4% underwent chemotherapy. These data are detailed in Table 2.

### 3.2. Neurological Symptoms and Factors Associated with Overall Survival

At the time of analysis, death occurred in 79 patients and 26 were still alive. The mOS from the BM diagnosis was 6.6 months (IQR: 5.1–9.2). The next step of the study explored the features of CM patients with symptomatic BMs (Table 3).

In our cohort, 45% of patients developed neurological symptoms. Their median age was 62 years and 57% were male. Many symptomatic patients showed supratentorial BMs (57%) as well as LDH levels ≤ 2 times the ULN (67%) at the time of their onset. Notably, only 47% of patients showed more than three BMs. The presence of neurological symptoms was a negative prognostic factor with a hazard ratio (HR) of 1.6 (95% CI 1.03–2.5). As shown in Figure 1A, mOS was 5.1 months for symptomatic patients versus 9.2 months for asymptomatic ones (*p* = 0.0354). Therefore, we questioned whether the eRT could have a role in the control of neurological symptoms, delaying clinical–neurological deterioration and then improving prognosis in symptomatic patients. In our analysis, patients with neurological symptoms showed a significant survival benefit from radiation therapy, achieving an mOS from BM diagnosis of 6.9 months with 2.9 months for those that did not receive eRT (*p* = 0.0234; Figure 1B). However, eRT also improved outcomes in asymptomatic patients (mOS: 11.8 vs. 2.7 months, *p* = 0.011; Figure 1C).

Other analyses demonstrated the poor prognostic role of LDH levels > 2 times the ULN at the time of encephalic progression. As shown in Figure 2A, by grouping patients by LDH levels we noticed a significant difference in terms of mOS (3.5 versus 9.2 months, *p* = 0.0014). Moreover, limited to the low number of patients, doubled LDH levels seem to select patients who do not benefit from eRT. Only patients with LDH levels ≤ 2 times the ULN levels showed an improvement in OS due to eRT. As shown in Figure 2B, OS was 2.5 months in the eRT untreated group with respect to 11.8 months observed in the group that received eRT (*p* < 0.0001). On the contrary, no survival difference was evidenced in patients showing LDH levels > 2 times the ULN after grouping them according to eRT (3.8 versus 3.5 months, *p* = 0.998; Figure 2C).

The Cox model applied to evaluate factors involved in survival showed that worse outcome was associated with the presence of more than three BMs (in the multivariable model: HR 1.78, 95% CI 1.04–3.06) and with LDH levels higher than 2-fold the ULN (HR 1.85, 95% CI 0.96–3.56). In addition, Table 4 shows that eRT has a protective effect (in the multivariable model: HR 0.37, 95% CI 0.21–0.67).

### 3.3. Impact of Treatments on Overall Survival

The next set of exploratory analyses investigated the additive effect of systemic therapies with eRT. As shown in Figure 3A,B, patients treated with TT or IT benefit from the concomitant use of eRT (mOS 2.2 vs. 9.5 months, *p* = 0.0062 and 2.7 vs. 9.9 months, *p* = 0.001, respectively). Moreover, the poor prognostic role of LDH levels was confirmed in patients treated with TT who experienced a worse outcome when LDH was >2 times the ULN (mOS 3.5 vs. 12.5 months, *p* = 0.0015; Figure 3C). On the contrary, patients who underwent IT did not show a significant difference in mOS when stratified according to LDH levels (mOS 4.17 vs. 8.43 months, *p* = 0.16; Figure 3D).

## 4. Discussion

The present study retrospectively explored the features of CM patients bearing BMs in a real-life cohort and examined the most used and effective multimodal therapeutic strategies in the pre-combined IT era or when clinical contraindications limit its practice.

In our population, BMs from CM occurred most frequently in males with a prevalent involvement of the supratentorial region. Most patients (90%) also evidenced extracranial metastases that in 63% of cases were diagnosed at least three months before the central nervous system (CNS) metastatic involvement. Thus, the intracranial progression apparently represents, at least in the vast majority of our patients, a delayed event in the natural history of CM. The aforementioned phenomenon has already been previously described, although a recent retrospective real-world study has assessed how BMs and extracranial metastases occur synchronously in nearly 70% of cases. This may be a potential consequence of intensive brain surveillance [34,35]. The incidence of *BRAF* mutations (55%) was similar to that reported in the general CM population. Nearly half of the patients (45%) suffered from neurological symptoms and, among them, only 46% showed more than three BMs, and only 24% were characterized by LDH levels > 2-fold the ULN. Therefore, neurological symptoms were not related to a multifocal metastatic brain disease or elevated LDH values.

The median survival after BM onset was 6.6 months, lower than the OS revealed by the recent Phase 3 randomized clinical trials using either TT or IT. However, our study reflects real-life data of patients usually excluded from clinical trials, such as those with poor performance status, previously treated with systemic therapies before the onset of BMs or those treated in the pre-combined IT era. In this setting, the prognosis is guided by factors mainly evidenced in retrospective cohorts of patients, including elevated LDH levels, neurological symptoms, three or more BMs and three or more extracranial metastatic sites [36]. In our population, the multivariate model was in line with these results.

Of note, neurological symptoms, occurring in 45% of our population, were the most common complication of BMs that negatively influenced both quality of life and survival. The percentage of symptomatic patients detected in our study differs from that evidenced in recent and quoted trials that highlight the high frequency of asymptomatic brain lesions (nearly 80%) [20,21]. In this regard, we have to underline that our study enrolled “real-life” patients who underwent radiological evaluation according to clinical practice with CT scans that have low specificity for BMs. Conversely, the gold standard for BM diagnosis is represented by the encephalic MRI, usually performed only after the evidence of neurological symptoms. Indeed, a retrospective study performed at the Memorial Sloan Kettering Cancer Center (MSKCC) extracted data from 355 real-life CM patients bearing BMs and found that 67% of them had neurological symptoms at BM onset [37]. Therefore, a universally accepted screening program, eventually involving encephalic MRI for high-risk patients, should be designed and diffused in clinical practice in order to detect BMs when the patients are still asymptomatic and improve the percentage of asymptomatic patients, as already happens in RCT.

Furthermore, we evaluated the eRT efficacy in improving outcomes across different subgroups of patients: symptomatic versus asymptomatic, IT-treated versus TT-treated and LDH levels upper versus lower than/equal to two times the ULN levels. The number of patients did not allow us to perform a subgroup analysis based on the different types of eRT. Anyway, this evaluation would not have significantly influenced any results considering that our aim was exploring the eRT additive effect with systemic treatments, its role in controlling neurological symptoms, improving quality of life and allowing access to further lines of systemic treatments, thus improving mOS. Our data showed that eRT improved outcomes both in the symptomatic and asymptomatic group as well as in patients treated with either TT or IT with a single agent. Thus, eRT might play a key role in prolonging OS by stabilizing BM growth and delaying clinical deterioration. Based on these results, while waiting for prospective data concerning the efficacy of eRT with combined IT, eRT should play a part in the therapeutic strategy at least for symptomatic brain metastatic CM patients or those who are asymptomatic and excluded from combined IT due to clinical contraindications.

The last step of our work focused on the negative prognostic role of LDH levels higher than 2-fold the ULN at the time of CNS metastatic spread and data parallel to those of previously published papers [38,39]. However, relevant data from our study concern the inefficacy of eRT in improving survival in patients with LDH > 2 times the ULN at BM onset. Thus, LDH values more than or equal to double the ULN could represent the sign of an aggressive and active brain metastatic disease that does not benefit from eRT despite the systemic treatments. On the other hand, the negative prognostic role of elevated LDH levels is confirmed only in patients treated with TT. Conversely, patients receiving IT with a single agent did not experience a significant difference in terms of OS when stratified according to LDH levels. The putative explanation of this finding could rely on a greater efficacy of IT in CM patients with BMs with LDH levels > 2-fold the ULN, confirming the recently published data regarding CM patients developing extracranial metastases [40]. The biological explanation of these findings relies on the role of LDH in downregulating the immune system due to the production of elevated levels of its oncometabolite, the lactate that was found to be associated with an increased number of metastatic sites and lower survival. Elevated levels of lactate induce an immune-suppressed microenvironment that sustains CM growth by promoting the expression of programmed cell death of protein-1 (PD-1) and the ligand (PD-L1) on tumor cells [41,42]. Once these data are confirmed in larger and prospective cohorts of CM patients bearing BMs, our results could underly that elevated LDH levels identify an aggressive subset of brain metastatic CM that should be oriented to IT-based therapy regardless of *BRAF* mutational status.

In conclusion, exploring the activity of the combined use of anti-CTLA4 and anti-PD1 agents in brain metastatic CM patients with LDH levels > 2-fold the ULN could represent an interesting strategy for the control of a severe complication that restrains survival in the majority of patients.

## 5. Conclusions

The present study demonstrated that neurological symptoms and high LDH values are negative prognostic factors in patients with CM developing BMs. In addition, preliminary results underlined the survival benefit due to the eRT in all subgroups, although patients with LDH levels ≥ 2-fold the ULN did not benefit from IT. These preliminary observations, therefore, suggest that IT may also have an active role in BMs showing elevated LDH levels. This may be at least explained by the immune suppressive microenvironment sustained by lactate, the LDH oncometabolite. However, further prospective studies are needed to understand the effective role of eRT and IT in melanoma patients characterized by BMs and elevated LDH.

## Figures and Tables

**Figure 1 cancers-15-01542-f001:**
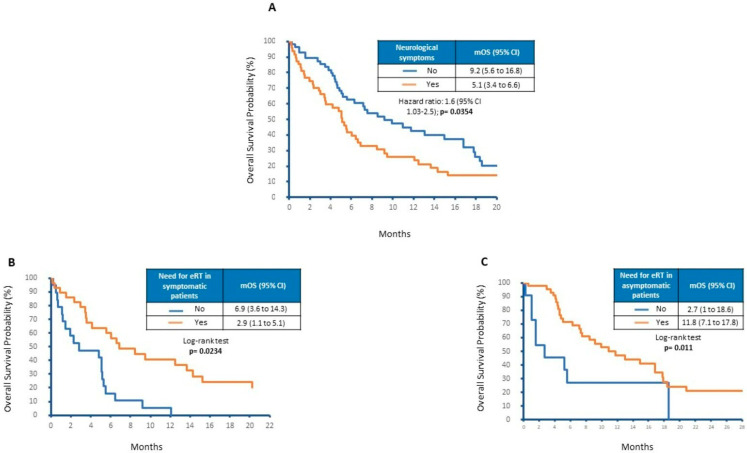
Overall survival according to (**A**) neurological symptoms, (**B**) eRT in symptomatic patients and (**C**) eRT in asymptomatic patients.

**Figure 2 cancers-15-01542-f002:**
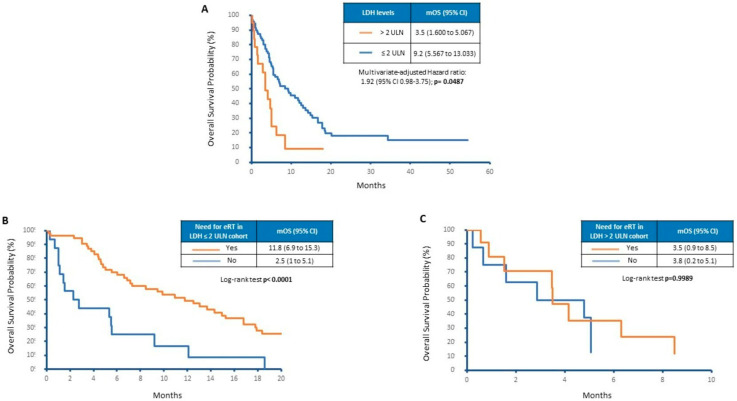
Overall survival of patients depending on (**A**) LDH levels, (**B**) eRT in LDH ≤ 2 ULN and (**C**) eRT in LDH > 2 ULN.

**Figure 3 cancers-15-01542-f003:**
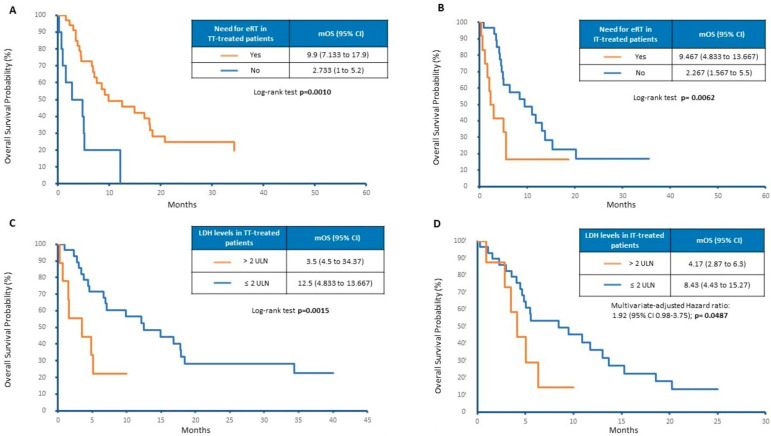
Overall survival by (**A**) eRT in TT-treated patients, (**B**) eRT in IT-treated patients, (**C**) LDH levels in TT-treated patients and (**D**) LDH levels in IT-treated patients.

**Table 1 cancers-15-01542-t001:** Patients’ demographics.

		Number	%
**Sex**	Male	61	58
	Female	44	42
**Primary site**	Limbs	36	34
	Head and neck	9	9
	Trunk	53	50
	Other sites	7	7
**Histology**	SSM	39	38
	Nodular	49	47
	Animal type	6	7
	Unknown	11	8
**Breslow depth**	≤1 mm	10	10
	>1–2 mm	11	11
	>2–4 mm	35	33
	>4 mm	34	32
	Unknown	15	14
**Ulceration**	Present	46	43
	Absent	50	48
	Unknown	9	9
**Mitosis**	≤1	38	36
	>1	67	64
**BRAF status**	Mutated	58	55
	Wild-type	47	45
**NRAS status**	Mutated	7	12
	Wild-type	53	88
**LDH at melanoma diagnosis**	≤2 × ULN	67	64
	>2 × ULN	5	5
	Unknown	33	31

SSM, superficial spreading melanoma; LDH, lactate dehydrogenase.

**Table 2 cancers-15-01542-t002:** Metastatic disease features.

		Number	%
**Extracranial metastases**	Yes	95	90
	No	10	10
**Number of extracranial metastatic sites**	≤2	52	50
	>2	37	35
	Unknown	16	15
**Timing of BMs from melanoma diagnosis**	Synchronous	3	3
	Metachronous	102	97
**Timing of BMs from extracranial progression**	Synchronous	38	36
	Subsequent	64	61
	Unknown	3	3
**Site of BMs**	Supra-tentorial	54	51
	Infra-tentorial	6	6
	Supra- and infra-tentorial	25	24
	Unknown	20	19
**Number of BMs**	≤3	62	59
	>3	39	37
	Unknown	20	19
**LDH levels at BM diagnosis**	≤2 × ULN	71	68
	>2 × ULN	19	18
	Unknown	15	14
**Symptomatic BMs**	Yes	47	45
	No	58	55
**Neurosurgery**	Yes	13	12
	No	92	88
**Encephalic radiotherapy**	Yes	75	71
	No	30	29
**Type of encephalic radiotherapy**	Stereotactic radiotherapy	49	65
	Whole brain radiotherapy	26	35
**First-line systemic treatment**	Targeted therapy with anti-BRAF + anti-MEK	43	41
	Immunotherapy	44	42
	Chemotherapy	3	3
	None	15	14

BM, brain metastasis; LDH, lactate dehydrogenase; ULN, upper limit normal.

**Table 3 cancers-15-01542-t003:** Features of symptomatic cohort.

		Number	%
**Sex**	Male	27	57
	Female	20	43
**BRAF status**	Mutated	26	55
	Wild-type	21	45
**LDH at BM diagnosis**	≤2 × ULN	31	67
	>2 × ULN	10	21
	Unknown	6	12
**Site of BMs**	Supra-tentorial	27	57
	Infra-tentorial	1	2
	Supra- and infra-tentorial	8	17
	Unknown	11	24
**Number of BMs**	≤3	25	53
	>3	22	47
**Steroid therapy**	Yes	47	100
	No	0	0

BM, brain metastasis; LDH, lactate dehydrogenase; ULN, upper limit normal.

**Table 4 cancers-15-01542-t004:** Results of Cox regression for overall survival.

Factors	Effect Tested	Univariate Analysis			Multivariate Analysis		
		HR	95% CI	*p*-Value	HR	95% CI	*p*-Value
**Sex**	Male vs. female	0.79	0.51–1.25	0.3161	0.92	0.54–0.57	0.7624
**Age (years)**	56–65 vs. ≤55	2.03	1.11–3.71	0.0215	1.68	0.88–3.22	0.1174
	66–75 vs. ≤55	1.59	0.86–2.92	0.1373	1.55	0.78–3.06	0.2070
	>75 vs. ≤55	2.28	1.1–4.7	0.0264	1.97	0.86–4.49	0.1080
**Radiotherapy**	Yes vs. no	0.3	0.18–0.49	<0.0001	0.37	0.21–0.67	0.0009
**Neurological symptoms**	Yes vs. no	1.6	1.03–2.49	0.0374	1.20	0.71–2.01	0.5020
**Number of BM**	≥3 vs. <3	1.45	0.93–2.28	0.571	1.78	1.04–3.06	0.0363
**Liver metastasis**	Yes vs. no	1.44	0.89–2.34	0.1375	1.05	0.59–1.85	0.8763
**LDH levels > 2 times the ULN**	Yes vs. no	2.58	1.41–4.74	0.0021	1.85	0.96–3.56	0.0452

BM, brain metastasis; LDH, lactate dehydrogenase; ULN, upper limit normal.

## Data Availability

The datasets used and analyzed during the current study are available from the corresponding author upon reasonable request.

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
