# Peer review of "Melanoma Brain Metastases: A Retrospective Analysis of Prognostic Factors and Efficacy of Multimodal Therapies"

_cancers, 2023, doi:10.3390/cancers15051542_

Round 1

Reviewer 1 Report

This interesting retrospective study showing the potential role of LDH as a predictive and prognostic biomarker in melanoma brain metastases. I recommend that authors re-run the Cox regression with liver metastases as a variable instead of the number of extracranial metastases. LDH elevation may be highly related to extensive liver involvement and, therefore worse prognosis due to that instead of the brain metastases

Author Response

Dear Reviewer,

Thanks for appreciating our work. We wish to thank the reviewer for his observations, so we have re-run the Cox model as suggested. We choose to not apply a selection method for covariates. The choice was made after evaluating the relationship between extracranial metastasis, liver metastasis, and LDH elevation. This analysis was performed by two separate logistic regressions with LDH elevation as a dependent and metastasis (liver or other sites) as a predictor. LDH elevation was not significantly associated either with liver metastasis or with extracranial metastasis. The logistic regression to evaluate the association between extracranial metastasis and liver metastasis, on the opposite, was statistically significant. Given the comment of the reviewer we have chosen to use the liver metastasis, so we could comment on the effect of the presence of liver metastases and the level of LDH, independently. As you can read, data seem not to change.

Reviewer 2 Report

The authors provide a nice real-life snapshot of factors influncing survival in patients with brain mets from melanoma in the pre-IT and TT era. Due to the retrospective design of their study, they should give more details on the modaities of patients' enrollment, how many hospitals were involved, was this a work only by medical oncologist and how many patients, if any, underwent neurosurgey. Also, they present discrepant results from what is expected on the timing of occurrence of brain mets as compared with recent and quoted studies that highlight high frequency of asymptomatic brain lesions. How do they explain this, is this perhaps due to differenes in the entity of brain surveillance in asymptomatic patients, please discuss

Author Response

Dear Reviewer,

Thanks for the suggestions.

- As you suggested we implemented the description of the study by specifying the number (2) and the name of the oncological centers when it was performed (lines 85-86). Also, we added that it was entirely conducted by medical oncologists (line 87).

- We also added the number of patients who underwent neurosurgery (13/105, 12%), lines 146-147. Moreover, we implemented table 2 with this information.

- As you suggested, the percentage of symptomatic patients evidenced in our work is higher than that found in the most relevant and famous randomized clinical trials (checkmate 2014 and combi-mb trial). We think that the reason relies on the “real-life” nature of our work and the timing of radiological evaluation with CT scan and not encephalic MRI performed in clinical practice. Indeed, we cited in the text another retrospective analysis performed at the Memorial Sloan-Kettering Cancer Center on 355 “real-life” patients with CM bearing BMs whom 65% of them evidenced neurological symptoms. We discussed these assumptions in lines 235-249

Round 2

Reviewer 2 Report

The authors have addressed the points raised